# Validation of reduced S-gene target performance and failure for rapid surveillance of SARS-CoV-2 variants

Cyndi Clark[1☯], Joshua Schrecker[1☯], Matthew Hardison[1], Michael S. Taitel[2]*

**1** Aegis Sciences Corporation, Nashville, TN, United States of America, **2** Walgreens, Deerfield, IL, United States of America

☯ These authors contributed equally to this work.
* Michael.taitel@walgreens.com

**Data Availability Statement:** All relevant data are within the paper and its Supporting Information files.

## Abstract

SARS-CoV-2, the virus that causes COVID-19, has many variants capable of rapid transmission causing serious illness. Timely surveillance of new variants is essential for an effective public health response. Ensuring availability and access to diagnostic and molecular testing is key to this type of surveillance. This study utilized reverse transcription polymerase chain reaction (RT-PCR) and whole genome sequencing results from COVID-19-positive patient samples obtained through a collaboration between Aegis Sciences Corporation and Walgreens Pharmacy that has conducted more than 8.5 million COVID-19 tests at ~5,200 locations across the United States and Puerto Rico. Viral evolution of SARS-CoV-2 can lead to mutations in the S-gene that cause reduced or failed S-gene amplification in diagnostic PCR tests. These anomalies, labeled reduced S-gene target performance (rSGTP) and S-gene target failure (SGTF), are characteristic of variants carrying the del69-70 mutation, such as Alpha and Omicron (B.1.1.529, BA.1, and BA.1.1) lineages. This observation has been validated by whole genome sequencing and can provide presumptive lineage data following completion of diagnostic PCR testing in 24–48 hours from collection. Active surveillance of trends in PCR and sequencing results is key to the identification of changes in viral transmission and emerging variants. This study shows that rSGTP and SGTF can be utilized for near real-time tracking and surveillance of SARS-CoV-2 variants, and is superior to the use of SGTF alone due to the significant proportion of Alpha and Omicron (B.1.1.529, BA.1, and BA.1.1) lineages known to carry the del69-70 mutation and observed to have S-gene amplification. Adopting new tools and techniques to both diagnose acute infections and expedite identification of emerging variants is critical to supporting public health.

## Introduction

Throughout the duration of the COVID-19 pandemic, mutations in the SARS-CoV-2 genome have led to the emergence of numerous variants. As of March 2022, over 1,900 unique lineages of SARS-CoV-2 have been identified via whole genome sequencing [1], a methodology that

**Funding:** The authors received no specific funding for this work.

**Competing interests:** The authors have declared that no competing interests exist.

has proven to be of great importance to public health surveillance during the pandemic [2]. Most new variants retain the same properties as their parent lineage, whereas others are classified as variants being monitored (VBM), variants of interest (VOI), variants of concern (VOC), or variants of high consequence (VOHC) based on an increased risk to global public health [3]. Since December 2020, there have been six VOCs: Alpha, Beta, Gamma, Delta, Epsilon, and Omicron based on data showing a reduced response to diagnostics, treatment, or vaccines, evidence of increased transmissibility, and/or demonstration of increased disease severity [4]. Subsequently, Alpha, Beta, Gamma, and Epsilon were reclassified as VBM amid the Delta surge in September of 2021. Omicron emerged in late November and has garnered much attention due to its significant increase in transmissibility and its expedient replacement of Delta as the predominant variant [5].

Surveillance data of circulating variants are typically available 2 weeks after the date of specimen collection due to the length of time required to complete whole genome sequencing. Efforts have been made to optimize sequencing methods and validate other methods to track the spread of viral lineages more efficiently. Some variants, such as Alpha and Omicron (B.1.1.529, BA.1, BA.1.1), have demonstrated unique result patterns during real-time reverse transcriptase polymerase chain reaction (RT-PCR) testing, allowing for rapid assessment of their presumptive presence in a patient specimen. When using the ThermoFisher TaqPath™ COVID-19 Combo Kit RT-PCR assay to detect N, ORF1ab, and S-genes, Alpha's and Omicron's mutated S-gene inhibits target amplification. This anomaly is labeled S-gene target failure (SGTF) [6] and is caused by deletions at positions 69 and 70 (del69-70) in the S protein [7]. Alpha, the dominant variant of concern in early 2021, shares the characteristic S-gene mutation (del69-70) with Omicron lineages B.1.1.529, BA.1, and BA.1.1. Surveillance of positive PCR results exhibiting SGTF allows for the identification of variants, such as Omicron, simultaneously with a positive PCR result within 24–48 hours; whereas strain identification to confirm lineage via SARS-CoV-2 whole genome sequencing can take up to 2 weeks after initial diagnostic testing.

Although the use of SGTF for rapid surveillance of viral spread has been reported in peer-reviewed literature, the rates of transmission are likely underreported based on analysis of SGTF alone [8]. This manuscript describes the validation of reduced S-gene target performance (rSGTP) analysis in addition to SGTF for early surveillance of variants with characteristic S-gene mutations, like del69-70. The use of additional measures for rapid surveillance of variants with rSGTP and SGTF allow for a real-time assessment of transmission and assist with identification of emerging variants during periods of transition.

Herein we show the use of a novel algorithm for rSGTP and SGTF based on the Ct values of the RT-PCR results as a proxy to rapidly assess the spread of certain SARS-CoV-2 lineages and validation of the algorithm with sequencing results in a randomized subset of specimens. With the emergence of BA.2, an Omicron lineage that does not share the del69-70 mutation or exhibit rSGTP or SGTF, we were also able to track the trajectory of B.1.1.529/BA.1/BA.1.1 versus BA.2 in near real-time using the algorithm and confirmed our analyses with sequencing data, demonstrating the utility of this tool during viral evolution and how this methodology may inform future surveillance measures.

## Methods

This study describes SARS-CoV-2 RT-PCR and whole genome sequencing results obtained via a collaboration between Aegis Sciences Corporation and Walgreens Pharmacy that has administered ~8.5 million COVID-19 tests at ~5,200 locations across the United States and Puerto Rico at the time of development of this manuscript [9]. RT-PCR testing utilizes primers and

probes that bind to regions along the viral genome and release fluorescent compounds through repeated cycles of gene target amplification. Samples tested via the ThermoFisher TaqPath™ COVID-19 Combo Kit are positive if the fluorescent signal is detected above threshold in two out of three gene targets, the positive control amplifies all three gene targets, and the negative control shows no amplification of the SARS-CoV-2 gene targets [10]. An internal control is included in each sample to account for extraction and PCR efficiency. SARS-CoV-2 lineages with characteristic S-gene mutations have altered amplification of the TaqPath™ S-gene target. This study analyzed rSGTP and SGTF for genome sequencing results meeting the following criteria:

- Reduced S-gene Target Performance (rSGTP):

  ○ Similar amplification of N and ORF1ab-genes, but reduced amplification of S-gene.

  ○ S-gene Ct value ≥ 4 + Average (N-gene Ct value: ORF1ab-gene Ct value)

- S-Gene Target Failure (SGTF):

  ○ Similar aAmplification of N and ORF1ab genes, but no amplification of the S-gene.

  ○ S-gene Ct value > 37 or null

These data were generated in a subset of COVID-19 positive specimens with an average N-gene and ORF1ab-gene Ct value ≤ 30 due to Ct fluctuations and poor sequencing results in samples with high Ct values and lower viral titers. Presumed Alpha and Omicron B.1.1.529/BA.1/BA.1.1 prevalence is calculated by taking the sum of rSGTP and SGTF RT-PCR positive results divided by the total number of RT-PCR positives for each collection date or a range of collection dates.

- Presumed Alpha or Omicron B.1.1.529/BA.1/BA.1.1% = [(rSGTP + SGTF) / TOTAL POSITIVES] * 100.

Presumed Omicron BA.2 prevalence is calculated by subtracting the sum of rSGTP and SGTF RT-PCR positive results from the total number of RT-PCR positives divided by the total number of RT-PCR positives for each collection date or a range of collection dates.

- Presumed Omicron BA.2% = [[TOTAL POSITIVES—(rSGTP + SGTF)] / TOTAL POSITIVES] * 100.

Whole genome sequencing (WGS) was performed using the Illumina® COVIDSeq Test and the NovaSeq 6000 sequencer. The Illumina® DRAGEN COVID Lineage App uses the FASTQ files to align reads to the SARS-CoV-2 reference genome (NC_045512.2) and reports coverage of targeted regions. Indeterminate (N) bases are assigned in regions covered with less than 20 unique reads or where the allele frequencies of A, T, G, or C are equal or too low to confidently make a call. Sequences with greater than 10% N bases were not included in the analysis. Variant calling, consensus sequence generation, and lineage/clade analysis is completed via Pangolin and NextClade pipelines.

Sensitivity, Specificity, Positive Predictive Value (PPV), and Negative Predictive Value (NPV) was determined for Presumed Alpha, Presumed Omicron B.1.1.529/BA.1/BA.1.1, and Presumed Omicron BA.2 calculations. The protocol for this study was institutional review board approved by the Pearl IRB™.

## Results

A randomized subset of COVID-19 positive samples collected between 1/1/2021–3/23/2022 with an average N-gene and ORF1ab-gene Ct value ≤ 30 (N = 374,469) were sequenced.

**Table 1. Rates of rSGTP and SGTF in WGS confirmed lineages.**

| Lineage Category | Total Samples Analyzed by Lineage | % of Total Samples Analyzed for Study | Total Samples with SGTF by Lineage | % of Samples with SGTF by Lineage | Total Samples with rSGTP by Lineage | % of Samples with rSGTP by Lineage | % of Samples with SGTF or rSGTP by Lineage |
|---|---|---|---|---|---|---|---|
| Alpha | 31,442 | 8.40% | 24,949 | 79.35% | 5,367 | 17.07% | 96.42% |
| Delta | 225,226 | 60.15% | 283 | 0.13% | 151 | 0.07% | 0.19% |
| Omicron B.1.1.529/ BA.1/ BA.1.1 | 79,829 | 23.32% | 73,583 | 92.18% | 5,960 | 7.47% | 99.65% |
| Omicron BA.2 | 3,737 | 1.00% | 11 | 0.29% | 2 | 0.05% | 0.35% |
| Other | 34,238 | 9.14% | 1,135 | 3.32% | 123 | 0.36% | 3.67% |

Lineages identified in these samples were: Alpha– 8.40% (n = 31,442), Delta– 60.15% (n = 225,226), Omicron B.1.1.529/BA.1/BA.1.1–21.32% (n = 79,826), Omicron BA.2 – 1.00% (n = 3,737) and Other– 9.14% (n = 34,238) (Table 1 and Fig 1). S-gene target failure (SGTF) was dominant in Alpha (79.35%) and Omicron B.1.1.529/BA.1/BA.1.1 (92.18%) but was not exhibited in all cases (Table 1 and Fig 1). Analysis of the non-SGTF samples that were classified as Alpha and Omicron B.1.1.529/BA.1/BA.1.1 showed lower amplification of the S-gene target when compared to the N-gene and ORF1ab-gene Ct values within the same sample.

Reduced S-gene target performance (rSGTP) was validated by comparing the frequency that the S-gene amplified and the distribution of the Ct value differences between the S-gene and the average of N and ORF1ab-genes for each sample where the lineage was confirmed as Alpha, Delta, Omicron, or Other (Table 2). The average Ct value of N and ORF1ab-genes was

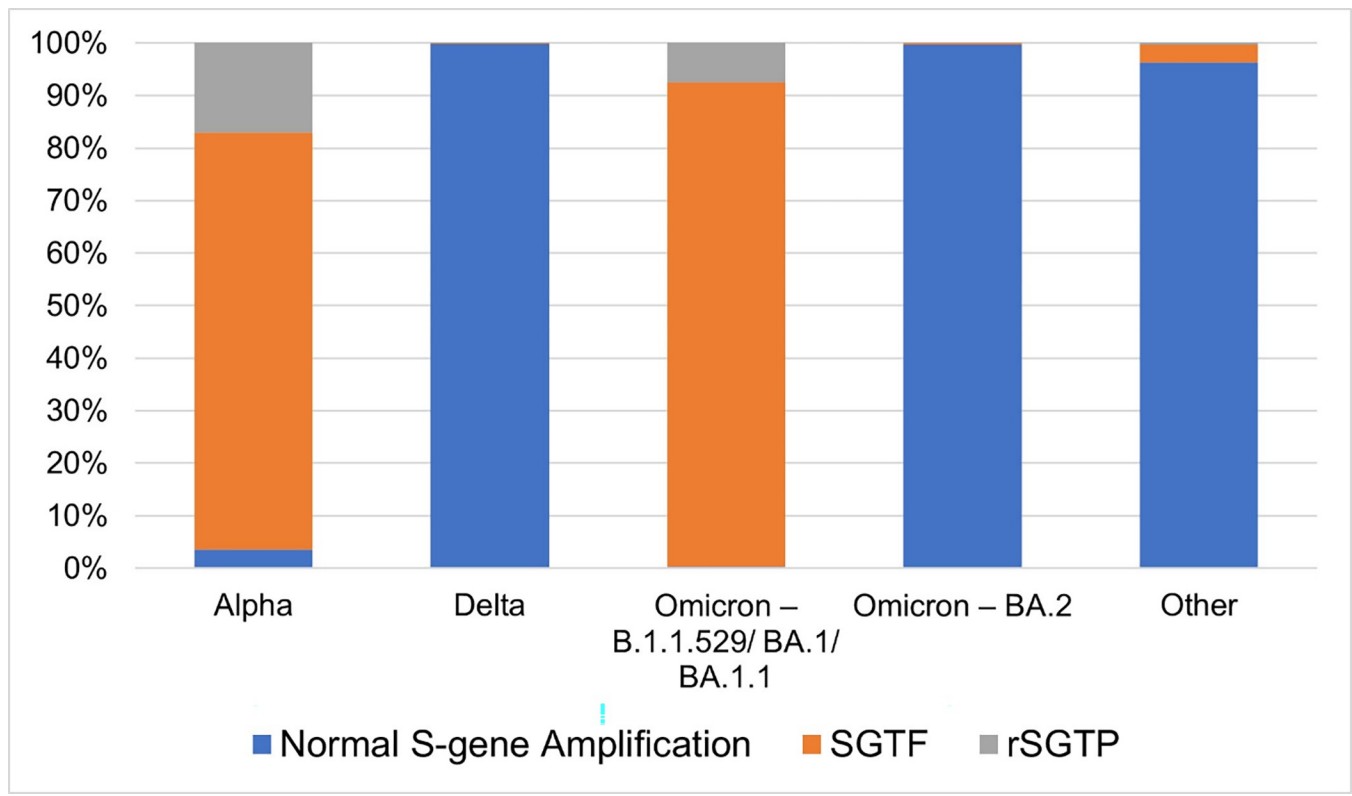

**Fig 1. Frequency of S-gene target status in WGS confirmed lineages.**

**Table 2. Ct value differences in WGS confirmed lineages with S-gene amplification.**

| Lineage Category | Total Samples with S-gene Present | Mean Ct Value Difference (S-gene Ct value–AVG Ct value (N: ORF1ab-genes)) | Median Ct Value Difference (S-gene Ct value–AVG Ct value (N: ORF1ab-genes)) | Mean Ct Value Difference (N gene Ct value–ORF1ab gene Ct value) | Median Ct Value Difference (N gene Ct value–ORF1ab gene Ct value) |
|---|---|---|---|---|---|
| Alpha | 6,493 | 5.08 | 5.31 | -0.23 | -0.17 |
| Delta | 224,943 | 0.36 | 0.28 | 0.25 | 0.30 |
| Omicron B.1.1.529/BA.1/BA.1.1 | 6,243 | 6.01 | 6.07 | 0.16 | 0.24 |
| Omicron BA.2 | 3,726 | 0.49 | 0.42 | 0.05 | 0.16 |
| Other | 33,103 | 0.23 | 0.19 | 0.38 | 0.39 |

chosen as the comparator because the mean and median Ct Value differences between N and ORF1ab-genes was less than 0.4 across all lineages analyzed (Table 2). The Ct value differences between the S-gene and the average of N and ORF1ab-genes amongst samples confirmed as Delta (Mean: 0.36, Median: 0.28), Omicron BA.2 (Mean: 0.49, Median: 0.42), or Other (Mean: 0.23, Median: 0.19) were negligible and the S-gene amplified in greater than 95% of cases. Conversely, S-gene amplification was observed in only 17.07% and 7.47% of Alpha and Omicron B.1.1.529/BA.1/BA.1.1 samples, respectively. Moreover, the Ct value of the S-gene was 3–4 cycles greater than the average Ct value of N and ORF1ab-genes in more than 90% of Alpha and Omicron B.1.1.529/BA.1/BA.1.1 cases in which S-gene amplification occurred (Fig 2). The S-gene amplified on average 5 cycles higher in Alpha and 6 cycles higher in Omicron B.1.1.529/BA.1/BA.1.1 samples demonstrating weak amplification, or reduced performance, of the S-gene target in samples where S-gene amplification occurred. (Fig 2). Using the algorithms for rSGTP: S-gene Ct value ≥ 4 + Average Ct value (N: ORF1ab) and SGTF: S-gene Ct value > 37, or null rSGTP + SGTF was observed in 96.42% of Alpha and 99.65% Omicron B.1.1.529/BA.1/BA.1.1 samples (Table 2).

Sensitivity and specificity of using rSGTP + SGTF or SGTF alone was calculated for Alpha, Omicron B.1.1.529/BA.1/BA.1.1, and Omicron BA.2 lineages that were confirmed by sequencing within the timeframe of first and last identification based on date of specimen collection (Table 3). Sensitivity improved by utilizing rSGTP + SGTF as a proxy for the identification of Alpha (96.4% vs. 79.3%) and Omicron B.1.1.529/BA.1/BA.1.1 (99.6% vs. 92.2%) in comparison to SGTF alone. Sensitivity did not change for the inverse of rSGTP + SGTF compared to the inverse of only SGTF in Omicron BA.2 (99.7%). Specificity remained consistent for both Alpha (99.5% vs. 99.6%) and Omicron B.1.1.529/BA.1/BA.1.1 (97.6% vs 97.8%) but increased for Omicron BA.2 (99.1 vs. 94.1%) when comparing rSGTP + SGTF to SGTF alone. The increase in sensitivity and negative predictive value (NPV) when rSGTP + SGTF is used as a proxy for surveillance of Alpha and Omicron B.1.1.529/BA.1/BA.1.1 lineages create a more accurate presumptive dataset with fewer false negatives. Furthermore, the utility of the inverse rSGTP + SGTF calculation is demonstrated by higher specificity and positive predictive value, or less false positives, in the presumed Omicron BA.2 dataset.

The prevalence of rSGTP + SGTF samples was graphed over time and compared to the prevalence of WGS-confirmed Alpha, Delta, Omicron B.1.1.529/BA.1/BA.1.1, and Omicron BA.2 (Fig 3). These data demonstrate the strong correlation between rSGTP + SGTF positives and the emergence and spread of both Alpha and Omicron B.1.1.529/BA.1/BA.1.1 VOC within the population and, for the first time, supports the use of reduced S-gene amplification (rSGTP) in samples as a proxy for early detection and surveillance of SARS-CoV-2 variants.

| Alpha | B.1.1.529/BA.1/BA.1.1 |
|---|---|

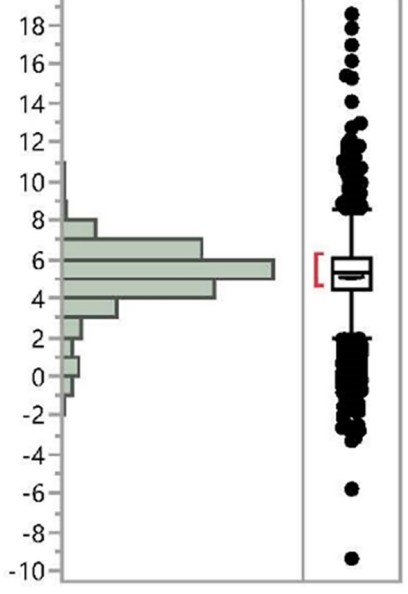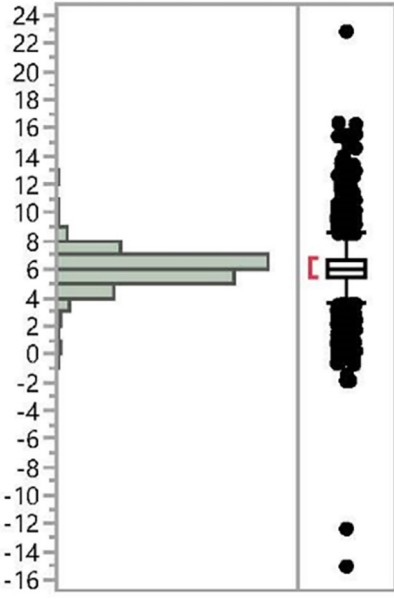

| Quantiles | | | | Quantiles | | |
|---|---|---|---|---|---|---|
| 100.0% | maximum | 18.57431425 | | 100.0% | maximum | 22.818301 |
| 99.5% | | 9.637467605 | | 99.5% | | 11.96101157 |
| 97.5% | | 7.6214769 | | 97.5% | | 8.3299015 |
| 90.0% | | 6.750757 | | 90.0% | | 7.2518295 |
| 75.0% | quartile | 6.0895925 | | 75.0% | quartile | 6.6525205 |
| 50.0% | median | 5.3118435 | | 50.0% | median | 6.070979 |
| 25.0% | quartile | 4.425855 | | 25.0% | quartile | 5.4155845 |
| 10.0% | | 3.1902082 | | 10.0% | | 4.6954773 |
| 2.5% | | 0.1716099125 | | 2.5% | | 3.22237485 |
| 0.5% | | -0.773169885 | | 0.5% | | 0.15341977 |
| 0.0% | minimum | -9.345805 | | 0.0% | minimum | -15.025473 |

| Summary Statistics | | | Summary Statistics | |
|---|---|---|---|---|
| Mean | 5.0755514 | | Mean | 6.0076117 |
| Std Dev | 1.7566684 | | Std Dev | 1.430093 |
| Std Err Mean | 0.0218005 | | Std Err Mean | 0.0180995 |
| Upper 95% Mean | 5.1182876 | | Upper 95% Mean | 6.0430931 |
| Lower 95% Mean | 5.0328152 | | Lower 95% Mean | 5.9721304 |
| N | 6493 | | N | 6243 |

**Fig 2. Distribution of Ct value differences between the S-gene and the average of N and ORF1ab-genes in Alpha and Omicron B.1.1.529/BA.1/BA.1.1 variants.**

**Table 3. Sensitivity, specificity, PPV, and NPV of proxy calculations by lineage.**

| Presumed Alpha | | | | |
|---|---|---|---|---|
| Alpha (Samples Collected: 1/1/2021-12/03/2021; n = 273,318) | Sensitivity | Specificity | PPV | NPV |
| SGTF | 79.3% | 99.6% | 96.5% | 97.4% |
| rSGTP + SGTF | 96.4% | 99.5% | 96.4% | 99.5% |
| Presumed Omicron B.1.1.529/BA.1/BA.1.1 | | | | |
| Omicron - B.1.1.529/BA.1/BA.1.1 (Samples Collected: 11/24/2021-3/23/2022; n = 105,646) | Sensitivity | Specificity | PPV | NPV |
| SGTF | 92.2% | 97.8% | 99.2% | 80.2% |
| rSGTP + SGTF | 99.6% | 97.6% | 99.2% | 98.9% |
| Presumed Omicron BA.2 | | | | |
| Omicron—BA.2 (Samples Collected: 1/3/2022-3/23/2022; n =: 59,920) | Sensitivity | Specificity | PPV | NPV |
| Inverse SGTF | 99.7% | 94.1% | 53.1% | 100.0% |
| Inverse rSGTP + SGTF | 99.7% | 99.1% | 88.0% | 100.0% |

## Discussion

SARS-CoV-2 genomes accumulate ~2 mutations per month, and over time, viral evolution gives rise to new lineages that are partitioned into phylogenetic trees and classified as variant strains [11]. Mutations in variants of concern (VOC), such as Omicron and previously Alpha, pose a challenge for some PCR primers to anneal to the S-gene. The phenomenon defined as reduced S-gene target performance (rSGTP) and failure (SGTF) is characteristic of both Alpha and Omicron B.1.1.529/BA.1/BA.1.1 variants. Our analyses revealed that although SGTF was present in the majority of cases confirmed as Alpha and Omicron B.1.1.529/BA.1/BA.1., there were many samples that had detectable S-gene amplification (Table 1). Identification of SGTF and rSGTP, instead of SGTF alone, can be used as a proxy to more accurately track emergence and spread of variants with the del69-70 mutation in real-time, enabling rapid dissemination

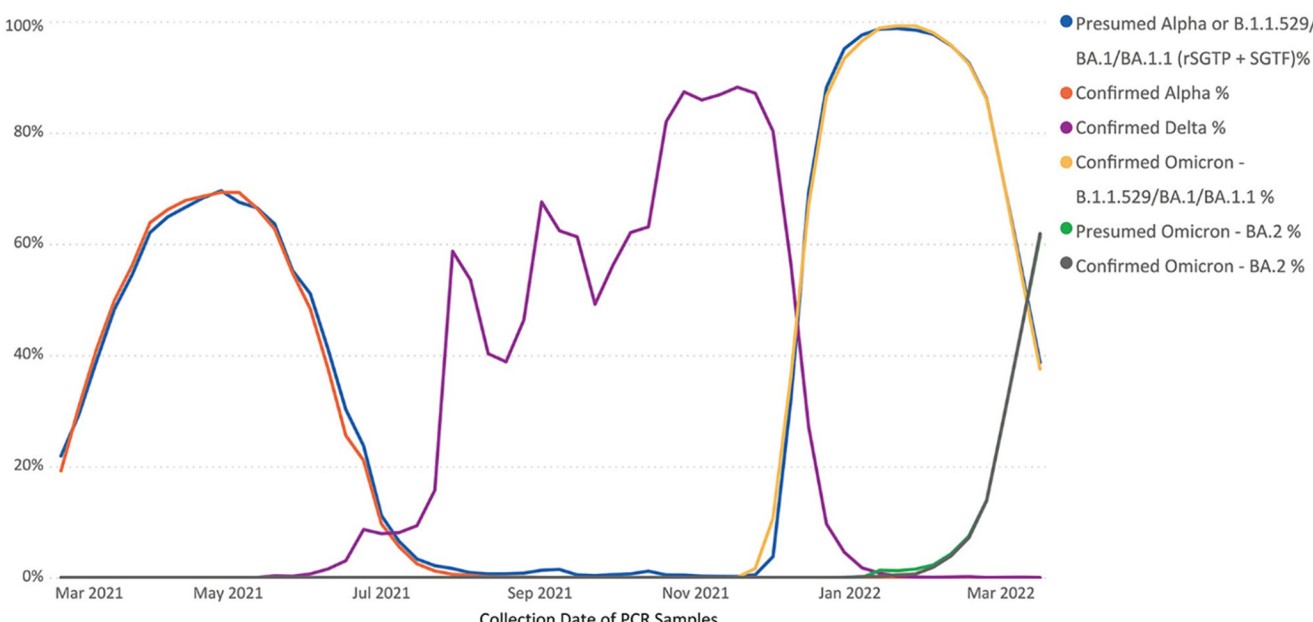

**Fig 3. Surveillance of SARS-CoV-2 variants via rSGTP + SGTF tracking versus WGS-lineage confirmation.** Date range truncated at March 2021 due to low sample volume for sequenced samples prior to this date.

of COVID-19 surveillance patterns. Additionally, the absence of rSGTP and SGTF has allowed for early assessment of transmission patterns of BA.2. Using the algorithm proposed for rSGTP and SGTF as a proxy for surveillance of Omicron and other variants of concern with these characteristic PCR result patterns, significantly accelerates the ability to identify gross changes in viral spread while waiting for viral genome sequencing to be completed.

Several previous publications have reported similar mechanisms for utilizing SGTF to increase efficiency of surveillance of certain variants of concern. To date, the publications have primarily focused on early detection of the Alpha and Omicron B.1.1.529/BA.1/BA.1.1. Investigators utilized similar methods for identification of SGTF as what is presented in this paper. This included identification of PCR results with amplification of the N and ORF1ab-genes in the absence of amplification for the S-gene. Similarly, other groups targeted PCR results with cycle threshold values $\leq 30$ for the N and ORF1ab-genes to reduce the incidence of identifying a sample as exhibiting SGTF in cases where viral load was low, and amplification was near the limit of the detection of the assay. These publications typically focused on smaller sample sizes but did confirm SGTF as a proxy for identification of the Alpha and B.1.1.529/BA.1/BA.1.1 variants through subsequent genome sequencing [8, 12–16]. Similar to our results, the vast majority of samples with SGTF, which were expected to have been caused by commonly circulating variants at the time of the study, were confirmed as Alpha or Omicron B.1.1.529/BA.1/BA.1.1 via sequencing. Additionally, we determined sensitivity (92.2%) and specificity (97.8%) was similar to previously published calculations utilizing SGTF to detect Omicron (B.1.1.529, BA.1, BA.1.1 lineages) [17]. Our analyses also revealed a significant proportion of Alpha and Omicron B.1.1.529/BA.1/BA.1.1 samples with S-gene amplification (Table 1). For this reason, we believe that utilization of SGTF alone for early surveillance of variants with del69-70 would underestimate the actual changes in transmission of these variants.

Migueres et al. reported a similar phenomenon that we identified in both Alpha and Omicron B.1.1.529/BA.1/BA.1.1 cases, which we refer to as rSGTP, though the analysis was performed only during a time at which Alpha was in circulation [18]. These investigators labeled samples with reduced S-gene amplification in comparison to that of N and ORF1ab-genes as "S-gene target late detection," or SGTL. More specifically, these samples were identified when S-gene Ct values were $\geq 5$ cycles higher than N or ORF1ab-genes and were reported as totaling less than 1% of all Alpha cases. Our analysis defined samples as having rSGTP when the S-gene cycle threshold value was at least 4 cycles higher than the average Ct values of N and ORF1ab, and nearly 17% of Alpha and 8% of Omicron B.1.1.529/BA.1/BA.1.1 cases exhibited rSGTP. Table 1 indicates that our calculation for the percent of samples with SGTF or rSGTP was unable to capture 100% of the confirmed Alpha (96.42%) or Omicron B.1.1.529/BA.1/BA.1.1 (99.65%) samples. The samples missed were those with Ct values for the S-gene less than 4 cycles higher than the average of N and ORF1ab-genes. Fig 2 demonstrates the distribution of the differences between the S-gene Ct value and the average of N and ORF1ab-genes Ct value in Alpha and Omicron B.1.1.529/BA.1/BA.1.1 lineages. According to poisson distribution, the calculation of rSGTP presented here captures ~90% or more of the Alpha and Omicron B.1.1.529/BA.1/BA.1.1 lineages exhibiting S-gene amplification and supports the utility and validity of using the definitions of SGTF and rSGTP outlined in this manuscript. Finally, we are aware that a research team at the CDC contemporaneously performed another analysis and came to similar conclusions regarding a revised SGTF definition (H. Scobie, Z. Smith, personal communication, April 13, 2022).

The primary limitation of this study is that all PCR testing for SARS-CoV-2 was performed utilizing the ThermoFisher TaqPath™ COVID-19 Combo Kit RT-PCR assay to detect N, ORF1ab, and S-genes. Although there would not be an expectation that other commercially available assays that include testing for the S-gene would have significantly different

capabilities in detecting amplification of this target, it is possible that results from testing performed by another assay may contribute to differences in the rates at which rSGTP and SGTF were identified. We did not perform testing via other methodologies to validate if rates of rSGTP and SGTF were similar across other assays that included testing for the S-gene.

An additional limitation of this study was its exclusion of test results from analysis with an average N-gene and ORF1ab-gene Ct value $\geq$ 30. These samples were excluded as those with moderate to low viral titer could have appeared to exhibit SGTF and rSGTP but may have had little to no detectable S-gene amplification due to proximity to the limit of detection of the assay. Additionally, samples with moderate to low viral titer can create limitations when performing whole genome sequencing as they often contribute to a significant increase in non-callable bases and result in less specific lineage calls [19]. The authors felt that inclusion of these low to moderate viral titer samples, which likely would have resulted in less specific lineage calls, would have negatively impacted the ability to compare findings between predominant lineages over time.

As we endure an ever-changing pandemic, the development of novel tools to both diagnose acute infections and to expedite identification of emerging variants will be critical during the transition into an endemic phase. Collaborative efforts between laboratories and healthcare providers to identify unique trends within large, shared datasets can provide support for public health decision making. Those engaged in protecting public health during the COVID-19 pandemic are urged to adopt new methods to identify and track this virus as it continues to mutate.

## Supporting information

**S1 File. Final COVID dataset full.** This Zip file contains the complete dataset used for all analyses, tables, and figures. Dates were adjusted by a random constant to assure deidentification. A data definition document is also included.
(ZIP)

## Acknowledgments

The authors thank Adam Wallace for his expert support preparing the data for this study and Heather Scobie and Zachary Smith for their review and feedback on this manuscript.

## Author Contributions

**Conceptualization:** Cyndi Clark, Joshua Schrecker, Michael S. Taitel.

**Data curation:** Cyndi Clark, Joshua Schrecker.

**Formal analysis:** Cyndi Clark, Joshua Schrecker.

**Funding acquisition:** Michael S. Taitel.

**Methodology:** Cyndi Clark.

**Project administration:** Joshua Schrecker, Michael S. Taitel.

**Resources:** Matthew Hardison.

**Supervision:** Matthew Hardison, Michael S. Taitel.

**Validation:** Cyndi Clark, Joshua Schrecker, Matthew Hardison, Michael S. Taitel.

**Visualization:** Joshua Schrecker, Michael S. Taitel.

**Writing – original draft:** Cyndi Clark, Joshua Schrecker, Michael S. Taitel.

**Writing – review & editing:** Cyndi Clark, Joshua Schrecker, Matthew Hardison, Michael S. Taitel.

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
