## [Decision Letter · Decision Letter 0]

8 Aug 2022

PONE-D-22-14356Validation of Reduced S-gene Target Performance and Failure for Rapid Surveillance of SARS-CoV-2 VariantsPLOS ONE

Dear Dr. Taitel,

Thank you for submitting your manuscript to PLOS ONE. After careful consideration, we feel that it has merit but does not fully meet PLOS ONE’s publication criteria as it currently stands. Therefore, we invite you to submit a revised version of the manuscript that addresses the points raised during the review process.

The study presents an important methodology and an algorithm to accurately survey for SARS-CoV-2 varaints of concern. However, reviewers and I have few questions and concerns which needs to be addressed. 

The general risk of this assay is in its inability to conclusively differentiate strains with the same deletion, as shown for alpha and omicron BA.1. however, it can differentiate variants with and without deletion mutant very effectively.

It is still not very clear to me how the S gene target performance reduction is measured. I understand you came up with a formula  but what is the comparator to dictate the performance is reduced? what if the primers/probe for S-gene are degraded or the sample has low viral load? how do you account for this? 

The following statement in discussion is very important in this study. I would recommend the authors to include this message concisely in the abstract as well "Our analyses also revealed a significant proportion of Alpha and Omicron B.1.1.529/BA.1/BA.1.1 samples with S-gene amplification (Table 1).For this reason, we believe that utilization of SGTF alone for early surveillance of variants with del69-70 would underestimate the actual changes in transmission of these variants".

please also highlight in the discussion, the limitation of the study and an explanation as in a scenario where any new emerging variants which does not carry the deletion at69-70 codon. in addition, please comment on the samples which fail due to the low viral load, which might be below the cut off of the assay, but might work fine for N and ORF1ab. 

you have briefly indicated that this was done with a specific RT PCR kit and in specific instrument. do you anticipate change in results due to change of qPCR instruments?

Please also add a comment on the descrepant samples, that were missed by the assay. 

We look forward to receiving your revised manuscript.

Kind regards,

Padmapriya P Banada, PhD

Academic Editor

PLOS ONE

Journal Requirements:

Additional Editor Comments:

The study presents an important methodology and an algorithm to accurately survey for SARS-CoV-2 varaints of concern. However, reviewers and I have few questions and concerns which needs tobe addressed.

The general risk of this assay is in the ability to not conclusively be able to differentiate strains with the same deletion, as shown for alpha and omicron BA.1. however, it can differentiate variants with and without deletion mutant very effectively.

It is still not very clear to me how the S gene target performance reduction is measured. I understand you came up with a formula, but what is the comparator to dictate the performance is reduced? what if the primers/probe for S-gene are degraded or the sample has low viral load? how do you account for this?

The following statement in discussion is very important in this study. I would recommend the authors to include this message concisely in the abstract as well "Our analyses also revealed a significant proportion of Alpha and Omicron B.1.1.529/BA.1/BA.1.1 samples with S-gene amplification (Table 1).For this reason, we believe that utilization of SGTF alone for early surveillance of variants with del69-70 would underestimate the actual changes in transmission of these variants".

please also highlight in the discussion, the limitation of the study and an explanation as in a scenario where any new emerging variants which does not carry the deletion at69-70 codon. in addition, please comment on the samples which fail due to the low viral load, which might be below the cut off of the assay, but might work fine for N and ORF1ab.

you have briefly indicated that this was done with a specific RT PCR kit and in specific instrument. do you anticipate change in results due to change of qPCR instruments?

Please also add a comment on the descrepant samples. that were missed by the assay.

Reviewers' comments:

Reviewer's Responses to Questions

**Comments to the Author**

1. Is the manuscript technically sound, and do the data support the conclusions?

Reviewer #1: Yes

Reviewer #2: Yes

2. Has the statistical analysis been performed appropriately and rigorously? 

Reviewer #1: Yes

Reviewer #2: I Don't Know

3. Have the authors made all data underlying the findings in their manuscript fully available?

Reviewer #1: Yes

Reviewer #2: Yes

4. Is the manuscript presented in an intelligible fashion and written in standard English?

Reviewer #1: Yes

Reviewer #2: Yes

5. Review Comments to the Author

Reviewer #1: General comments:

Clark et al. presents an interesting study using Ct differences between different targets to identify reduced S-gene target performance instead of solely S-gene target failure to increase sensitivity in the preliminary identification of SARS-CoV-2 variants for rapid surveillance purposes. This proof-of-concept approach to increase screening sensitivity could be applied if new VOCs with different target failure or reduction patterns emerge.

Line numbers would be useful for reviewing. Further clarification about the exact methods in the process would helpful.

Introduction

1. Page 2: Suggest aligning classification of VBM, VOI, VOC, VOHC with WHO classifications instead of CDC to make it internationally applicable.

2. Page 2: Suggest taking out Epsilon in the list of VOCs, as has not been listed as a VOC by the WHO and did not have worldwide impact.

Methods

3. Page 4: rSGTP criteria second point about S-gene Ct value >4 + Average (N-gene Ct value: ORF1ab-gene Ct value) is unclear. Could be rephrased clearer with wording instead of using “+” Perhaps S-gene Ct value 4 greater than the Average etc.

4. Page 4: Was there a Ct value threshold for the amplification of N and ORF1ab genes in the two criteria? Would include the threshold that is used for transparency.

5. Page 5-6: Methods section about sensitivity, specificity, PPV and NPV - would detail what the gold standard comparison was to identify true positives and negatives versus false. I assume it was to the sequences that underwent NGS. Were all samples first identified via the algorithmic method with RT-PCR screening and then 1:1 subsequently underwent gold standard NGS? Or was it a sample that went for NGS that was not connected 1:1?

Results

6. Page 6: First line of the second results paragraph may need to be rephrased for clarity.

7. Page 7: Would again rephrase the rSGTP criteria for clarity. Perhaps using the word Ct value 4 greater than the average etc.

8. Page 7: Unclear what is meant by inverse rSGTP + SGTF in the second paragraph. Recommend clarifying.

9. Page 7 & Figure 3: Visually quite striking in terms of correlation between presumed lineage vs confirmed lineage. Was any statistical testing done beyond visual with the statement of “strong correlation”?

10. Page 7 & Figure 3: Were the confirmed variants the same sample that was screened or was it all that was sequenced (ie a larger sample)?

Discussion

11. Page 8: First paragraph, last sentence – may want to reframe it less as a comparison between this algorithm vs WGS and rather this algorithm with rSGTP compared to just SGTF alone adds to the fidelity and confidence of using this screening approach in more timely surveillance. As the authors have stated, the SGTF approach is already being implemented as a timely surveillance method.

12. Page 10: Can see another limitation being samples with low viral load/high cycle Cts. Would be interesting to see as a percentage of overall samples. How many fit that criteria and was excluded from this validation study and therefore affect the sensitivity and correlation if looking at all samples?

13. Could the authors comment about this approach in the context of the CDC framework for evaluating surveillance systems? This will strengthen the paper in connecting it into the larger surveillance discussion instead of focused more on the laboratory side of things.

Figures

13. Table 1: For the column % of total samples analyzed for study – is there a reason it totals to 102%? I see the % for Omicron is different from the manuscript versus the table. Typo? Recommend putting a footnote of why that is if it’s not a typo.

14. Figure 1: If keeping this figure, consider putting in percentages if possible for the different stacked bars.

15. Table 3: Would recommend having a 2x2 table with the actual numbers of true positives, true negatives, false positives, and false negatives in addition to percentages for more clarity.

16. Figure 3: Suggest labeling the y-axis. May be a point to clarify is this all samples that underwent any type of SGTP screening or the subset samples that was part of this study as per previous points.

Reviewer #2: This was a good validation study. Please review paper for grammatical errors.

Page 4 & 5: Remove bullets. The criteria for rSGTP and SGTF can be written as sentences. The formulas can be submitted in a supplementary table.

Page 6:

• Table 1 states that 79,829 Omicron B.1.1.529/BA.1/BA.1.1 samples were analyzed, but paragraph one states 79,826 Omicron B.1.1.529/BA.1/BA.1.1 samples were sequenced. What happened to the remaining 3 Omicron samples?

• The last sentence in paragraph one might work better in the rSGTP paragraph.

Page 7:

• Sentence 1: Remove the criteria for rSGTP and SGTF. It’s stated in your method section.

6. PLOS authors have the option to publish the peer review history of their article (what does this mean?). If published, this will include your full peer review and any attached files.

Reviewer #1: No

Reviewer #2: No

---

## [Author Response · Author response to Decision Letter 0]

24 Aug 2022

Response to Reviewers,

Thank you for the thorough review of our manuscript and the opportunity to submit a revised version. We have addressed each of the comments below and updated the manuscript where necessary. 

• The general risk of this assay is in its inability to conclusively differentiate strains with the same deletion, as shown for Alpha and Omicron BA.1. however, it can differentiate variants with and without deletion mutants very effectively.

o Response: This is a limitation of the approach that we have addressed in the manuscript discussion.

• It is still not very clear to me how the S-gene target performance reduction is measured. I understand you came up with a formula but what is the comparator to dictate the performance is reduced? 

o Response: The comparator to dictate the S-gene target performance is reduced is the performance of the N and ORF1ab gene targets within the same sample. The performance of all targets is measured by the amplification Ct value. There was a negligible difference between the Ct values of the N and ORF1ab-genes across all samples analyzed. The Mean and Median Ct Value difference between N and ORF1ab-genes for the lineages in this study has now been added to Table 2.

The Mean and Median Ct Value difference between N and ORF1ab-genes is less than 0.4 across all lineages. Once this was determined, we used the average Ct value of these two targets to compare against the S-gene Ct value. 

The Mean and Median Ct Value difference between the S-gene and the average of N and ORF1ab-genes is less than 0.5 in Delta, Omicron BA.2, and Other lineages, indicating similar S-gene target amplification, or performance, in these lineages.

The Mean and Median Ct Value difference between the S-gene and the average of N and ORF1ab-genes is 5.08 and 5.31 for Alpha and 6.01 and 6.07 for Omicron B.1.1.529/BA.1/BA.1.1 lineages, respectively. The higher S-gene Ct Values in this sample set demonstrates weak amplification, or reduced performance, of the assay target in those samples where S-gene amplification occurred. 

o Response: Additionally, Table 1 and Figure 1 compares the frequency of rSGTP across all confirmed lineages in this study and further validates rSGTP in lineages with del69-70.

The proportion of samples with rSGTP was 0.07%. 0.05%, and 0.36% in Delta, Omicron BA.2, and Other lineages, respectively.

The proportion of samples with rSGTP was 17.07% and 7.47% in Alpha and Omicron B.1.1.529/BA.1/BA.1.1 lineages, respectively

• What if the primers/probe for S-gene are degraded or the sample has low viral load? How do you account for this?

o Response: Degradation of the primers and probes would be observed in the positive controls. There are four positive controls on each RT-PCR run and all three genes, S, N, and ORF1ab, must amplify for the run and samples to pass. This has been added to the methods.

o Response: To address samples with low viral loads, we specifically analyzed samples with an average N-gene and ORF1ab-gene Ct value < 30 to reduce the bias potentially introduced by these samples. Additionally, samples with lower viral load typically do not meet our quality criteria, or that which was set forth by the CDC, to make a definitive lineage call via our next-generation sequencing platform. These samples either result in incomplete sequencing, or result in a call for a less specific parent lineage. This is likely why the ”Other” category has an occurrence of SGTF and rSGTP that is higher than that found with Delta and BA.2. This is addressed in the methods and has been added as a limitation to the manuscript.

• The following statement in the discussion is very important in this study. I would recommend the authors to include this message concisely in the abstract as well "Our analyses also revealed a significant proportion of Alpha and Omicron B.1.1.529/BA.1/BA.1.1 samples with S-gene amplification (Table 1). For this reason, we believe that utilization of SGTF alone for early surveillance of variants with del69-70 would underestimate the actual changes in transmission of these variants".

o Response: This has been concisely added to the abstract and manuscript.

• Please also highlight in the discussion, the limitation of the study and an explanation as in a scenario where any new emerging variants which does not carry the deletion at 69-70 codon. In addition, please comment on the samples which fail due to the low viral load, which might be below the cut off of the assay, but might work fine for N and ORF1ab.

o Response: Statement of clarification and limitation added to the manuscript discussion. 

• You have briefly indicated that this was done with a specific RT-PCR kit and in specific instrument. Do you anticipate change in results due to change of qPCR instruments?

o Response: No meaningful changes would be expected due to a change of qPCR instrumentation. There could potentially be slight changes in the Ct values associated with each sample, but this would be minimally impactful and within the normal expected variance range of repeated analysis using this molecular technique. Additionally, other labs using the same reagent kit with different qPCR instruments have also observed SGTF and rSGTP in these lineages. 

• Please also add a comment on the discrepant samples, that were missed by the assay. 

o Response: Table 1 indicates that our calculation for % of Samples with SGTF or rSGTP was unable to capture 100% of the confirmed Alpha (96.42%) or Omicron B.1.1.529/BA.1/BA.1.1 (99.65%) samples. The samples that were missed by our calculation were those where the Ct value of the S-gene was less than 4 cycles higher than the average of N and ORF1ab-gene targets. Figure 2 demonstrates the distribution of the differences between the S-gene Ct value and the average of N and ORF1ab-genes Ct value in Alpha and Omicron B.1.1.529/BA.1/BA.1.1 lineages. According to Poisson distribution, our calculation of rSGTP would capture ~90% or more of the Alpha and Omicron B.1.1.529/BA.1/BA.1.1 lineages with reduced S-gene target performance. This has been clarified in the results and discussion sections.

---

## [Editor Report · Decision Letter 1]

12 Sep 2022

Validation of Reduced S-gene Target Performance and Failure for Rapid Surveillance of SARS-CoV-2 Variants

PONE-D-22-14356R1

Dear Dr. Taitel,

We’re pleased to inform you that your manuscript has been judged scientifically suitable for publication and will be formally accepted for publication once it meets all outstanding technical requirements.

Kind regards,

Padmapriya P Banada, PhD

Academic Editor

PLOS ONE
---

## [Editor Report · Acceptance letter]

21 Sep 2022

PONE-D-22-14356R1 

Validation of Reduced S-gene Target Performance and Failure for Rapid Surveillance of SARS-CoV-2 Variants 

Dear Dr. Taitel:

I'm pleased to inform you that your manuscript has been deemed suitable for publication in PLOS ONE. Congratulations! Your manuscript is now with our production department. 

Kind regards, 

on behalf of

Dr. Padmapriya P Banada 

Academic Editor

PLOS ONE